 # INSTRUCTCELL: A Multimodal Cell Language Model for Single-cell Analysis

## Author Name
email@example.com

## Abstract

As Large Language Models (LLMs) rapidly evolve, their influence in science is becoming increasingly prominent. The emerging capabilities of LLMs in task generalization and free-form dialogue can significantly advance fields like chemistry and biology. However, the field of single-cell biology, which forms the foundational building blocks of living organisms, still faces several challenges. High knowledge barriers and limited scalability in current methods restrict the full exploitation of LLMs in mastering single-cell data, impeding direct accessibility and rapid iteration. To this end, we introduce INSTRUCTCELL, which signifies a paradigm shift by facilitating single-cell analysis with natural language. By thoroughly understanding single-cell instructions through the multimodal architecture, INSTRUCTCELL has acquired profound expertise in single-cell biology and the capability to accommodate a diverse range of analysis tasks. Extensive experiments further demonstrate INSTRUCTCELL's robust performance and potential to deepen single-cell insights, paving the way for more accessible and intuitive exploration in this pivotal field.

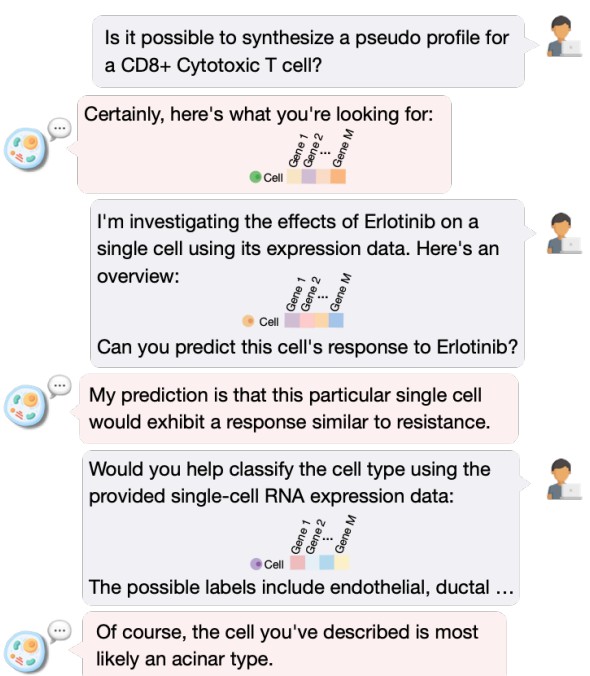

Figure 1: INSTRUCTCELL facilitates single-cell analysis through conversational interactions.

## 1 Introduction

Artificial Intelligence for Science (AI4Science) has emerged as a pivotal force in advancing scientific research [Wang *et al.*, 2023; Tinn *et al.*, 2023], particularly in complex domains like nuclear fusion [Degrave *et al.*, 2022], protein structure prediction [Jumper *et al.*, 2021], and autonomous chemical discovery [Daniil *et al.*, 2023]. Among the various AI tools, Large Language Models (LLMs) are at the forefront, demonstrating significant advancements in fields such as biology and chemistry [Zheng *et al.*, 2023; Zhao *et al.*, 2023; Zhang *et al.*, 2024]. These models excel in interpreting biological sequential data and following human instructions [Tong and Zhang, 2023; Fang *et al.*, 2024], making human language an essential medium for acquiring biological insights. As a result, LLMs are breaking down barriers to biological knowledge, revolutionizing research paradigms, and deepening our understanding of life sciences.

This paradigm shift opens new avenues for single-cell biology research, a field pivotal to understanding the basic units of life. Single-cell biology examines the intricate functions of these cells, ranging from energy production to genetic information transfer [Bechtel, 2006], playing a critical role in unraveling the fundamental principles of life and mechanisms influencing health and disease [Pollard *et al.*, 2022]. The field has witnessed a surge in single-cell RNA sequencing (scRNA-seq) data, driven by advancements in high-throughput sequencing and reduced costs. Repositories like the Gene Expression Omnibus (GEO) [Barrett *et al.*, 2012] and the Human Cell Atlas (HCA) [Regev *et al.*, 2017] have been instrumental in accumulating and disseminating this data. The emerging field of single-cell foundation models, such as scBERT and scGPT [Yang *et al.*, 2022; Theodoris *et al.*, 2023; Cui *et al.*, 2023; Mo *et al.*, 2021], is changing traditional task-specific approaches [Lieberman *et*

*al.*, 2018; Shao *et al.*, 2021; Liao *et al.*, 2022]. These models leverage extensive scRNA-seq datasets, applying NLP techniques to analyze gene expression matrices—structured formats that simplify scRNA-seq data into computationally tractable representations—during pre-training. They are subsequently fine-tuned for distinct single-cell analysis tasks. Despite their potential, the technical intricacies and knowledge prerequisites of these models pose challenges to their accessibility and practical application, especially in fast-paced iteration scenarios.

Recent efforts have been directed towards adapting LLMs for critical single-cell analysis tasks. For example, using ChatGPT for cell type annotation [Hou and Ji, 2023], converting cells into sequences of gene names and fine-tuning LLMs for single-cell analysis tasks [Levine *et al.*, 2023], and retrieving textual summaries of genes from the NCBI gene database followed by obtaining gene embeddings through GPT-3.5 [Chen and Zou, 2023]. However, text and cells represent two fundamentally different forms of language, with distinct representation spaces and sequence semantics. Textual language is an abstraction based on human linguistic expression, while scRNA-seq profiles the expression pattern of each gene within a cell. Treating these as a single modality can lead to information loss and hinder the model's ability to deeply understand and master the connections between them.

In this study, we introduce INSTRUCTCELL, a multimodal cell language model that leverages natural language to enhance single-cell analysis. Initially, we construct a single-cell instruction dataset that LLMs can readily interpret. Subsequently, by employing a multimodal architecture, INSTRUCTCELL is designed to handle both high-dimensional cellular data and structured textual data, effectively merging quantitative cell expression profiles with qualitative textual annotations. To enhance the LLM's expertise in the single-cell domain, we conduct instruction tuning on single-cell instructions to adeptly execute a range of single-cell tasks. INSTRUCTCELL leverages a unique encoding strategy where cellular data and textual data are co-encoded into a shared latent space. This allows for the direct comparison and combination of genomic information with textual descriptions, facilitating a more detailed understanding of cellular functions. Moreover, INSTRUCTCELL enables researchers to input human instructions, thereby facilitating the convenient execution of essential tasks in single-cell analysis.

## 2 Related Work

**Single-cell analysis.** Single-cell analysis delves into the examination and manipulation of individual cells, aiming to decipher their specific roles in complex biological systems. This discipline leverages scRNA-seq to reveal the active genes and their expression levels within single cells [Plass *et al.*, 2018; Cao *et al.*, 2019]. For efficient analysis, scRNA-seq data is organized into gene expression matrices, where columns and rows correspond to individual cells and genes, respectively, and the matrix values reflect gene expression levels [Brazma and Vilo, 2000]. Utilizing these matrices enables researchers to handle a range of critical tasks in single-cell analysis, such as dissecting the cellular composition of tissues and identifying novel cell types and states. The challenges in this field, including managing high-dimensional data [Wu and Zhang, 2020; Tejada-Lapuerta *et al.*, 2023], addressing data sparsity [Bouland *et al.*, 2023], and handling the computational demands of large-scale data analysis [Wolf *et al.*, 2018], are being addressed by the development of innovative computational tools and algorithms. These advancements are crucial for distilling reliable and biologically relevant insights from single-cell data.

**Single-cell foundation models.** Initial attempts to analyze gene expression matrics involve machine learning methods and autoencoder-based approaches [Liu *et al.*, 2021; Oller-Moreno *et al.*, 2021; Ji *et al.*, 2021]. However, these studies often produce models tailored for specific tasks, which lack the adaptability for broader analytical applications [Angerer *et al.*, 2017]. Inspired by the success of foundation models in NLP tasks [Devlin *et al.*, 2019; Lewis *et al.*, 2020], the concept is naturally extended to the single-cell domain. Single-cell foundation models emerge to offer wide-ranging capabilities across various single-cell analysis tasks. ScBERT [Yang *et al.*, 2022] acquires insights into individual and combined gene expressions by analyzing millions of normalized scRNA-seq data within the BERT framework. Geneformer [Theodoris *et al.*, 2023] employs a self-supervised masked token prediction objective to decode gene networks, subsequently fine-tuning for chromatin and network dynamics tasks. ScGPT [Cui *et al.*, 2023] benefits from generative pre-training, excelling in functions like cell type annotation, gene perturbation prediction, and pseudo-cell generation. Distinct from foundation models, INSTRUCTCELL employs instruction tuning on single-cell instructions, equipping the model with the ability to accurately follow instructions across various single-cell analysis tasks without the need for pre-training and fine-tuning.

**Instruction-following models.** The inherent strength of LLMs lies in their ability to follow and execute human instructions. Trained on specialized instruction datasets, these models develop a deep understanding of intricate instructions, offering flexibility and a broader scope compared to traditional foundation models. This versatility has led to diverse innovations within biology, such as language-guided molecular design [Edwards *et al.*, 2022; Fang *et al.*, 2024; Zeng *et al.*, 2023], medical question-answering [Singhal *et al.*, 2023], and automated experimental design [Daniil *et al.*, 2023]. The exploration of instruction-following models is emerging as a promising avenue in the single-cell domain. GPTCelltype [Hou and Ji, 2023] explores the feasibility of using GPT-4 for cell type annotation, indicating a new step forward in language-guided single-cell analysis. Cell2sentence [Levine *et al.*, 2023] demonstrates how gene expression profiles can be translated into gene name sequences, illustrating the potential for integrating LLMs into analyzing single-cell data. Apart from them, INSTRUCTCELL employs a multimodal architecture to familiarize LLMs with scRNA-seq data and extend their proficiency across a variety of tasks through instruction tuning. It facilitates a seamless entry for researchers into the field, allowing direct information acquisition through chat and thereby enhancing the ac-

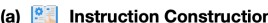

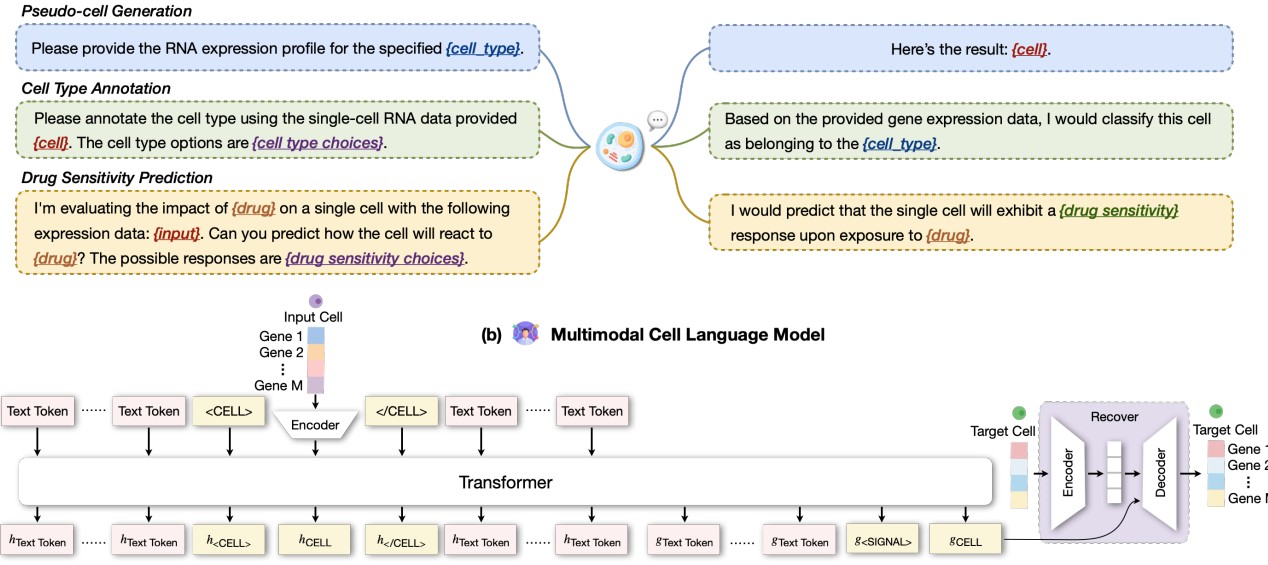

Figure 2: The overview of INSTRUCTCELL.

cessibility of single-cell analysis.

## 3  Methodology

### 3.1  Instruction Construction

The objective of INSTRUCTCELL is to enable researchers to conduct comprehensive single-cell analysis using natural language inputs, ensuring LLM's adeptness in both single-cell and natural language. For this purpose, we construct a single-cell instruction dataset. We collect scRNA-seq data from publicly available single-cell datasets and design templates corresponding to different tasks, transforming them into instructions for LLMs to understand. These instructions can be in the form of pure text or a mixture of text and scRNA-seq data. As illustrated in Figure 2 (a), we focus on the following single-cell tasks outlined:

***Pseudo-cell Generation.*** Pseudo-cell generation focuses on generating gene expression profile tailored to specific cell type labels. The prompt requests the model to construct a cell for a given cell type, and the target is expected to be a cell accurately representing the gene expression profile of that cell type. This task is vital for unraveling gene expression and regulation across different cell types, offering insights for medical research and disease studies, particularly in the context of diseased cell types.

***Cell Type Annotation.*** For cell type annotation, the model is tasked with precisely classifying cells into their respective types based on gene expressions. Here, the prompt involves providing a gene expression profile for the model to determine the cell type, with the target being the accurate identification and classification of that cell type. This task is fundamental for understanding cellular functions and interactions within tissues and organs, playing a crucial role in developmental biology and regenerative medicine.

***Drug Sensitivity Prediction.*** The drug sensitivity prediction task aims to predict the response of different cells to various drugs. In this task, the prompt presents a cell along with a specific drug, and the model is tasked with predicting the cell's response to the drug. The target is an accurate prediction of the cell's sensitivity or resistance to the drug. It is pivotal in designing effective, personalized treatment plans and contributes significantly to drug development, especially in optimizing drug efficacy and safety.

In real-world scenarios, human communication exhibits inherent diversity and complexity, characterized by a wide array of linguistic styles and expressions. INSTRUCTCELL, designed to engage in conversational interactions, must be adept at handling this linguistic variability. For each task, we start with a clear and concise human-written description. This description is then fed into GPT-4, leveraging its capability to produce diverse renditions of the same concept. This diversity in training ensures that INSTRUCTCELL learns to understand and respond to different modes of language expression, making it a robust tool for versatile communicative interactions in single-cell analysis.

### 3.2  Multimodal Cell Language Model

In order to enable the model to simultaneously handle both text and single-cell data modalities, INSTRUCTCELL is built on a multimodal language model architecture that facilitates cross-modal knowledge sharing, enhancing the model's ability to process different data types. As illustrated in Figure 2 (b), tokens or embeddings related to text and cells are represented in pink and yellow, respectively.

For precise processing of instructions containing single-cell gene expression data, we designed special symbols <CELL> and </CELL> to mark the beginning and end of cell data. This strategy allows the model to semantically differentiate between text and single-cell data, preventing confusion

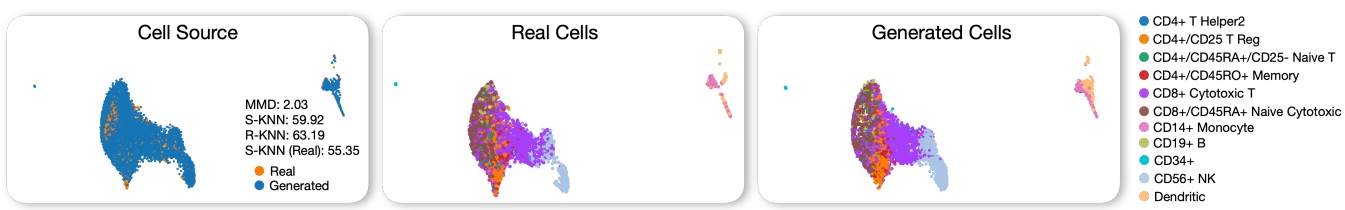

Figure 3: Distribution of real cells and generated cells.

between data types. The single-cell gene expression data is mapped to a cell latent representation by a dedicated encoder and then is fed into the embedding layer along with text data for combined encoding. This design enables the model to understand both textual information and cell features in the same semantic space, enhancing the model's comprehensive expressive capabilities.

In tasks that generate purely textual data, such as cell type annotation and drug sensitivity prediction, we treat these as sequence generation tasks. This method leverages the model's pre-trained language understanding capabilities to generate relevant textual outputs, improving the relevance and accuracy of the outputs. In Figure 2 (b), the symbols of $h_{\text{Text Token}}$ represent embeddings from the pre-trained model, while $g_{\text{Text Token}}$ symbols are newly generated embeddings that will be mapped back to specific tokens to produce the final textual output.

The pseudo-cell generation task is more complex, requiring the model to generate single-cell gene expression profiles. To facilitate this, we first introduce a $g_{\texttt{<SIGNAL>}}$ to remind the language model that it will next generate the cell's embedding, $g_{\text{CELL}}$. This signaling mechanism helps maintain direction and accuracy in the generation task. We then employ an autoencoder module to reconstruct the model-generated $g_{\text{CELL}}$ into a single-cell gene expression profile. It is important to note that during the inference phase, we remove the encoder in this module and retain only the decoder. This architectural choice enables INSTRUCTCELL not only to handle complex multimodal inputs but also to provide high-quality predictions and analyses across various bioinformatics and medical applications. By exposing this multimodal cell language model to a wide variety of single-cell analysis tasks, it not only identifies task-specific patterns but also develops a holistic understanding of the entire domain.

## 4 Experiments

### 4.1 Experimental Settings

***Baselines.*** The compared baselines include scGPT [Cui *et al.*, 2023], and scBERT [Yang *et al.*, 2022], each specifically designed for single-cell analysis.

***Implementation Details.*** INSTRUCTCELL is implemented using the PyTorch framework and trained on 4 Nvidia V100 GPUs. We initialize our model using T5-base model [Raffel *et al.*, 2020] as pre-trained foundations. For the pseudo-cell generation task, we utilize 68,185 fresh peripheral blood mononuclear cells (PBMCs) from the PBMC68K dataset [Zheng *et al.*, 2017]. For the cell type annotation task, we select and clean cells from two datasets: 2,108 cells

from the pancreas (pancreatic islet) sequenced using Smart-seq2 [Segerstolpe *et al.*, 2016], and 20,528 cells from the pancreas (pancreatic islet) sequenced using SMARTer [Xin *et al.*, 2016]. For the drug sensitivity prediction task, we select two datasets: GSE149383 [Aissa *et al.*, 2021] records the response of 2,254 human lung cancer cells to the drug Erlotinib, and GSE117872 [Sharma *et al.*, 2018; Ravasio *et al.*, 2020; Suphavilai *et al.*, 2021] records the response of 1,302 human oral squamous cancer cells to the drug Cisplatin. All these instructions are split into train/validation/test datasets in an 8:1:1 ratio.

### 4.2 Pseudo-cell Generation

When given specific cell types, we evaluate the performance of the generated cells using the following metrics:

- *MMD (Maximum Mean Discrepancy)*: Used to measure if the differences between the model-generated samples and the actual samples are sufficiently small. A smaller MMD value indicates better performance of the model in simulating single-cell data.

- *S-KNN (Self-KNN)*: This custom metric assesses whether the model-generated single cells possess biological significance. Specifically, for a simulated single-cell dataset, assume category $i$ contains $c_i$ cells; the model needs to generate $c_i$ cells for each category. After generating the cells, the system calculates the nearest $K$ neighbors for each cell, excluding the cell itself, and evaluates the label consistency based on these neighbors' labels. The average label consistency across all cells is computed and defined as the SKNN metric.

- *R-KNN (Real-KNN)*: We also introduce the R-KNN metric to ensure that not only do the model-generated single cells have biological meaning, but this biological significance is consistent with that of actual cells. Specifically, we use the cells in the test set as the training set for a KNN classifier and the model-generated single cells as the test set. We then calculate the accuracy of the KNN classifier, which defines the R-KNN metric.

To assess the biological accuracy of gene expression profile generated by INSTRUCTCELL, we visualize their corresponding gene expression matrices in two dimensions, following the pseudo-cell generation experimental setting. Figure 3 shows that cell distributions generated by INSTRUCTCELL closely resemble those of real cells, confirmed by the low MMD. The distinct clustering of cell types in the generated data demonstrates INSTRUCTCELL's ability to capture and differentiate unique cell characteristics, indicating its capacity to generate detailed profiles for each cell type.

| Model | Segerstolpe-2016 | | | Xin-2016 | | |
|---|---|---|---|---|---|---|
| | ⇅ Accuracy | ⇅ Average F1 | ⇅ Weighted F1 | ⇅ Accuracy | ⇅ Average F1 | ⇅ Weighted F1 |
| scBERT | 96.42 | 96.38 | 95.44 | 98.57 | 98.22 | 98.01 |
| scGPT | 97.51 | 97.51 | 97.56 | 98.90 | 98.90 | 98.91 |
| **INSTRUCTCELL** | **99.50** | **99.17** | **99.50** | **99.50** | **99.39** | **99.50** |

Table 1: Performance (%) of cell type annotation on two datasets.

| Model | GSE149383 | | | GSE117872 | | |
|---|---|---|---|---|---|---|
| | ⇅ Accuracy | ⇅ Average F1 | ⇅ Weighted F1 | ⇅ Accuracy | ⇅ Average F1 | ⇅ Weighted F1 |
| scBERT | **97.82** | **97.82** | **97.52** | 93.13 | 93.58 | 93.10 |
| scGPT | 96.46 | 96.46 | 96.45 | 80.15 | 80.15 | 80.94 |
| **INSTRUCTCELL** | 97.35 | 97.34 | 97.35 | **95.42** | **95.52** | **95.42** |

Table 2: Performance (%) of drug sensitivity prediction on two datasets.

## 4.3 Cell Type Annotation

INSTRUCTCELL revolutionizes cell type annotation by eliminating the need for classifier training. Instead, task descriptions and gene expression profiles are fed directly as instructions into the model, which then predicts the cell type through a sequence generation manner. Table 1 reveals that INSTRUCTCELL holds a distinct edge over competing models, reflecting its proficiency in deciphering complex relationships between gene expressions and corresponding cell types. The effectiveness of INSTRUCTCELL in this context is enhanced by its ability to process and integrate verbal instructions. These instructions not only specify the task at hand but also contextually enrich the model's analysis, providing a linguistic framework that aligns with biological data. This integration of language and biology through a multimodal lens allows INSTRUCTCELL to extract more meaningful insights from the data, leading to more accurate predictions.

## 4.4 Drug Sensitivity Prediction

Similarly, the task of drug sensitivity prediction is also accomplished in an autoregressive manner, obviating the need for a distinct classifier. As shown in Table 2, experiments are conducted on two drug response datasets. INSTRUCTCELL outperforms in both the GSE149383 and GSE117872 datasets, surpassing the single-cell foundation model scGPT and achieving performance levels comparable to the single-cell foundation model scBERT. This performance advantage stems from our model's ability to contextually analyze and synthesize information across different modalities. Rather than treating textual and cellular data as separate entities, INSTRUCTCELL interprets them as complementary sources of information, leading to more accurate and robust predictions.

## 5 Conclusion and Future Work

In this work, we propose INSTRUCTCELL, a multimodal cell language model that facilitates single-cell analysis with natural language. This approach not only bridges the gap between disparate data modalities but also ingeniously integrates them, enhancing the LLM's capability to process and interpret complex biological data. Our study on INSTRUCTCELL confirms its proficiency in deciphering complex single-cell data and its versatility across a wide range of analysis tasks. Interesting future directions include: *i)* integrating more single-cell analysis tasks, *ii)* applying INSTRUCTCELL in personalized medicine to tailor drug treatments.

## Ethical Statement

There are no ethical issues.

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
