# OpenReview forum: "InstructCell: A Multimodal Cell Language Model for Single-cell Analysis"
_ijcai.org/IJCAI/2024/Workshop/AI4Research — AI4Research 2024_

### Official Review · Reviewer_5Z86 · 2024-05-22

**Rating:** 3
**Confidence:** 5

**Review:**

This paper proposes an instruct-based method to construct multimodal LLM for single-cell transcriptomic data analysis.

I think the authors ignored many papers related to this topic [1,2,3,4] with similar structure, and the motivation of using "multimodal LLM" is very confusing. Normally, multimodal data in single-cell level represent different biological features from one cell, like DNA, protein, etc. The authors should consider improving their method under such direction.

[1] Levine, Daniel, et al. "Cell2sentence: Teaching large language models the language of biology." bioRxiv (2023): 2023-09.

[2] Fang, Yin, et al. "ChatCell: Facilitating Single-Cell Analysis with Natural Language." arXiv preprint arXiv:2402.08303 (2024).

[3] CELLama: Foundation Model for Single Cell and Spatial Transcriptomics by Cell Embedding Leveraging Language Model Abilities

[4] Theodoris, Christina V., et al. "Transfer learning enables predictions in network biology." Nature 618.7965 (2023): 616-624.

---

### Official Review · Reviewer_z2vJ · 2024-05-30
**Review Summary**

**Rating:** 6
**Confidence:** 4

**Review:**

Firstly, I am pleased to see the efforts made by Large Language Models (LLMs) in the field of single-cell RNA-seq analysis. The article is well-organized and demonstrates a clear line of thought, making significant contributions towards making single-cell analysis more accessible and intuitive. Here are some specific comments on the article:

1. **Overview of INSTRUCTCELL**: The overview of INSTRUCTCELL is too simplistic and does not provide sufficient details to understand how the INSTRUCTCELL model accomplishes the tasks of “Pseudo-cell Generation,” “Cell Type Annotation,” and “Drug Sensitivity Prediction.”

2. **Construction of Multimodal Language Model**: The description of the method used to construct the multimodal language model in INSTRUCTCELL is overly brief. A more detailed explanation of the process is necessary to fully appreciate the approach.

3. **Visualization of Results**: While presenting the results for cell type annotation and drug sensitivity prediction, the use of only tables is insufficient. It would be more effective to include visual output results along with the tables to provide a clearer and more intuitive understanding.

4. **Comparison with Conventional Methods**: When comparing the functionalities of INSTRUCTCELL, it would be more impactful to also compare it with conventional single-cell analysis methods. Visual comparisons would enhance the understanding and effectiveness of the comparison.

5. **Open Source Code**: Making the model code open source would be highly beneficial. This would facilitate further research and application, and allow other researchers to replicate and build upon this work.

By addressing these points, the article could provide a more comprehensive and detailed understanding of INSTRUCTCELL’s capabilities and its advantages over existing methods.

---

### Official Review · Reviewer_nhfc · 2024-06-01
**InstructCell is a promising and user-friendly LLM for single-cell analysis tasks**

**Rating:** 8
**Confidence:** 4

**Review:**

Authors developed a multimodal cell language model called InstructCell that is able to perform three tasks (pseudo-cell generation, cell type annotation and drug sensitivity prediction) through using natural languages and/or single-cell sequencing data as inputs. I think this paper has an overall high quality, with well-structured sections for introduction, methodology, experiments and conclusions. I especially like the figures and tables, which provide clear visual support for the textual content.

Pros:

1. InstructCell integrates natural language and scRNA-seq data as multimodal inputs, making single-cell analysis more accessible to users with limited technical background.

2. InstructCell employs instruction tuning to adapt the model for various single-cell tasks, eliminating the need for pre-training and fine-tuning of most foundation models.

3. The paper has a clear and detailed description of the model structure and experimental design, facilitating reproducibility.

Cons:

1. Although the paper evaluates the model on all three tasks, the benchmark is not very comprehensive. For example, the pseudo-cell generation task is assessed only qualitatively through visualizations. Additionally, the cell type annotation performance is evaluated exclusively on pancreas data, which is relatively easier to annotate compared to other tissue types.

2. The instructions on how to use InstructCell are not provided in great detail.

3. While the selected tasks are important in the field, including additional single-cell analysis tasks or single-cell multi-model data would further validate the model's versatility and attract more future users.

---

### Decision · Program_Chairs · 2024-06-03

Accept